# Beyond Muscle Weakness: Unraveling Endocrine and Metabolic Dysfunctions in Duchenne Muscular Dystrophy, a Narrative Review

**DOI:** 10.3390/biomedicines13071613

**Published:** 2025-07-01

**Authors:** Giuseppe Cannalire, Giacomo Biasucci, Vanessa Sambati, Tommaso Toschetti, Arianna Maria Bellani, Anna-Mariia Shulhai, Federica Casadei, Erika Rita Di Bari, Francesca Ferraboschi, Cecilia Parenti, Maria Carmela Pera, Susanna Esposito, Maria Elisabeth Street

**Affiliations:** 1Pediatrics and Neonatology Unit, University of Parma, Guglielmo da Saliceto Hospital, 29121 Piacenza, Italy; g.cannalire@ausl.pc.it; 2Department of Medicine and Surgery, University of Parma, 43126 Parma, Italy; vanessa.sambati@unipr.it (V.S.); tommaso.toschetti@gmail.com (T.T.); ariannamaria.bellani@unipr.it (A.M.B.); annashulhai@gmail.com (A.-M.S.); federica.casadei@unipr.it (F.C.); erikarita.dibari@unipr.it (E.R.D.B.); francesca.ferraboschi@unipr.it (F.F.); cecilia.parenti@unipr.it (C.P.); mariacarmela.pera@unipr.it (M.C.P.); susannamariaroberta.esposito@unipr.it (S.E.); mariaelisabeth.street@unipr.it (M.E.S.); 3Pediatric Clinic, University Hospital of Parma, 43126 Parma, Italy

**Keywords:** Duchenne muscular dystrophy, growth impairment, endocrine dysfunction, metabolic alterations, osteoporosis, puberty delay, RANK/RANKL/OPG pathway, GHD, GH treatment, obesity, insulin resistance

## Abstract

**Background:** Duchenne muscular dystrophy (DMD) is a severe X-linked neuromuscular disorder caused by mutations in the *DMD* gene, leading to progressive muscle degeneration, loss of ambulation, and multi-systemic complications. Beyond its impact on mobility, DMD is associated with significant endocrine and metabolic dysfunctions that develop over time. **Objective:** To provide a comprehensive analysis of growth disturbances, endocrine dysfunctions, and metabolic complications in DMD including bone metabolism, considering the underlying mechanisms, clinical implications, and management strategies for daily clinical guidance. **Methods:** In this narrative review, an evaluation of the literature was conducted by searching the Medline database via the PubMed, Scopus, and Web of Science interfaces. **Results:** Growth retardation is a hallmark feature of DMD, with patients exhibiting significantly shorter stature compared to their healthy peers. This is exacerbated by long-term glucocorticoid therapy, which disrupts the growth hormone/insulin-like growth factor-1 (GH/IGF-1) axis and delays puberty. Obesity prevalence follows a biphasic trend, with increased risk in early disease stages due to reduced mobility and corticosteroid use, followed by a decline in body mass index (BMI) in later stages due to muscle wasting. Metabolic complications, including insulin resistance, altered lipid metabolism, and hepatic steatosis, further characterize disease burden. Osteoporosis and increased fracture risk, primarily due to reduced mechanical loading and glucocorticoid-induced bone resorption, are major concerns, needing early screening and intervention. The RANK/RANKL/OPG signaling pathway has emerged as a critical factor in bone deterioration, providing potential therapeutic targets for improving skeletal health. **Conclusions:** Growth and endocrine disorders in DMD are complex and multifactorial, requiring proactive monitoring and early intervention. Addressing these issues requires a multidisciplinary approach integrating endocrine, nutritional, and bone health management. Further research is essential to refine treatment strategies that mitigate growth and metabolic disturbances while preserving overall patient well-being.

## 1. Introduction

Duchenne muscular dystrophy (DMD) is a severe *X-linked* neuromuscular disorder characterized by progressive skeletal muscle degeneration, loss of independent ambulation typically by the early teenage years, and multi-systemic complications that collectively shorten life expectancy. DMD presents in children’s early life and rapidly evolves, leading to significant disability. Its prevalence at birth is estimated between 15.9 and 20 cases per 100,000 male live births and in the general population varies from 1.7 to 8.3 cases per 100,000 individuals [1,2,3,4]. DMD does not vary among races or ethnic groups [5]. Prevalence variations among countries can be ascribed to different research methodologies, diagnostic criteria, and genetic variations. Advancements in treatment have contributed to increased life expectancy, allowing achieving a third and fourth decade of life even if with severe disability [2]. DMD is a lethal *X-linked* recessive neuromuscular disorder caused by mutations in the *DMD* gene that result in absent or insufficient functional dystrophin, a cytoskeletal protein that enables the strength, stability, and functionality of myofibers [6]. Mutations in the *DMD* gene can also cause Becker muscular dystrophy (BMD), which is a milder disease with a later onset and a slower progression than DMD [7].

DMD presents many challenges in terms of diagnosis, management, and treatment development. Early and accurate diagnosis remains difficult because of the variability in disease onset and progression, which often leads to delays in therapeutic interventions [6]. Despite advancements in corticosteroid therapy and supportive care, the disease remains incurable with limited treatment options available to stop or significantly slow down progression [7]. Currently, approved disease-modifying therapies include ataluren (for nonsense-mutation DMD), exon-skipping agents such as eteplirsen, golodirsen, and viltolarsen (for selected exon deletions), and the corticosteroid vamorolone, which offers a more favorable safety profile compared to prednisone and deflazacort. DMD is associated with a relevant economic burden, with both direct and indirect costs related to long-term care and complications [8]. Furthermore, gene therapy, cell-based treatments, and exon-skipping approaches show promising results but face significant problems, including immune responses, limited efficacy, and accessibility concerns [9]. The role of microRNAs in disease pathogenesis is also under investigation, offering new potential therapeutic targets. Finally, inflammation and immune system interactions play a crucial role in muscle degeneration, adding complexity for disease-modifying therapies [10] and causing thyroid tissue fibrosis leading to hypothyroidism [11]. Addressing these challenges requires a multidisciplinary approach, combining molecular research, innovative therapeutics, and comprehensive patient care strategies.

The diagnosis of DMD is based on clinical features, laboratory tests, and genetic analyses. Early clinical signs include progressive muscle weakness with walking difficulty, frequent falls, and a possible family history of the disease [7]. The main biochemical marker is represented by serum creatine phosphokinase (CPK), which can increase by 10–50 times above normal levels [6]. Diagnostic confirmation consists of genetic analyses that identify mutations in the DMD gene. A muscle biopsy may be performed if genetic tests are inconclusive, confirming dystrophin absence through immunohistochemical techniques [6,8]. Diagnostic tests also involve electrocardiogram and cardiac ultrasound to assess possible signs of cardiomyopathy [5].

The inherited myogenic illnesses known as muscular dystrophies are characterized by various stages of progressive muscle wasting and weakening. In line with the distribution of major muscle weakness, they can be categorized into a number of groups, including congenital forms (Duchenne, Becker, and Emery–Dreifuss), distal, facioscapulohumeral, oculopharyngeal, and the most heterogeneous group, limb–girdle. The different forms of the condition differ significantly in terms of severity, age at presentation, rate of advancement, ensuing consequences, and prognosis [9]. In DMD, symptoms present early, around 3 years, causing loss of ambulation within 13 years of life, followed by cardiac complications, such as dilated cardiomyopathy and arrhythmia, and respiratory disorders, including chronic respiratory failure. In the first phase of the disease, the child experiences frequent difficulty in running, climbing stairs, jumping, getting up from the ground, falls, and develops a waddling gait with a positive “Gowers’ sign” [1]. Worldwide, most children are diagnosed around 5 years of age, when their physical abilities diverge considerably from those of their peers; if muscle strength deteriorates, boys require the use of a wheelchair before adolescence [8]. The subsequent impairment of the cardiac and respiratory systems is the main cause of death for these patients [1]. Most patients become wheelchair-dependent around 10–12 years of age and need assisted ventilation at around 20 years of age [7]. Nevertheless, survival has greatly improved over time thanks to the use of nocturnal ventilation and spinal surgery, with 30% of patients surviving beyond 30 years of age [1]. The general symptoms of DMD are reported in Figure 1.

DMD is caused by mutations in the DMD gene that is located on the *X chromosome (Xp21.2-p21.1)*, which encodes for dystrophin, a large cytoskeletal protein essential for maintaining the structural integrity of muscle fibers (Figure 2). One of dystrophin’s essential roles is to bind directly to filamentous actin (F-actin) in the cytoskeleton and anchor this to the dystrophin-associated protein complex (DAPC), which spans from the sarcolemma to the extracellular matrix via laminin. The absence or marked deficiency of functional dystrophin destabilizes the DAPC, leading to increased sarcolemmal fragility and a heightened susceptibility to contraction-induced micro-ruptures. This results in uncontrolled influx of calcium ions (Ca^2+^), which activates calcium-dependent proteases, causes mitochondrial dysfunction, and subsequent activation of apoptosis. In addition, mechanical stress and myofiber damage lead to chronic inflammation mediated by macrophages and proinflammatory cytokines, further exacerbating muscle degeneration. Repeated cycles of degeneration and insufficient regeneration contribute to the progressive replacement of muscle tissue by adipose tissue and fat [7]. The pathophysiology of DMD is reported in Figure 2.

## 2. Materials and Methods

In this narrative review, we retrieved the more recent studies evaluating epidemiology, challenges in DMD, main features (including inflammation and knowledge from basic science), and natural history. New therapeutic strategies were also searched. Search criteria for studies in the literature were human studies, full-text papers, English language, and as order of choice systematic reviews, metanalyses, and randomized controlled trials and observational studies if the above were missing. Narrative reviews were considered when available to verify that all the literature had been included. The time elapsed from publication date (<10 years) and age (from birth to 18 years). Studies were evaluated as the most significant according to number of cases, multicenter studies, and importance of results for human health. Additional relevant studies were obtained by screening and searching references and papers published before 2015 that were cited in the text for their importance within the review’s aim.

We searched PubMed, Scopus, and Web of Science, on 30 January 2025, with targeted keywords: Duchenne muscular dystrophy, epidemiology, quality of life, growth, growth deficiency, growth hormone, metabolic changes, metabolism, glucose metabolism, insulin, obesity, dyslipidemia, bone metabolism, bone mineral density, osteoporosis, endocrine disorders, thyroid, thyroiditis glucocorticoids, mesangioblast stem cell therapy, treatment, new treatments, clinical trials. Official sites of the WHO and European international organizations were also reviewed.

The search was performed using a string adding (AND) the terms postnatal growth OR growth OR growth hormone treatment OR short stature OR puberty OR testosterone OR bisphosphonates OR bone mineral density OR osteoporosis OR fractures OR bone loss OR muscle dysfunction OR myokines OR glucocorticoid OR bisphosphonates OR anti-RANKL OR cell therapy OR gene therapy OR CRISPR-Cas OR regenerative medicine OR myogenic stem cells OR Givinostat to previous search terms. We then performed a hand-screening of all the reference lists included in the papers to discover studies missed in the primary search process. Conference abstracts and qualitative studies (i.e., interviews, letters) were discarded. All authors scored retrieved titles and abstracts independently. Subsequently, full texts of all potentially relevant papers were accurately examined and included only if they reported useful information describing the actual quality of life and the impact of new treatment strategies.

## 3. Results and Discussion

A synthesis of the results was provided based on each identified theme.

### 3.1. The Role of Glucocorticoids in Duchenne Muscular Dystrophy (DMD)

Glucocorticoids (GCs) remain a cornerstone in the management of DMD as they improve muscle strength, motor function, and delay disease progression. The most prescribed corticosteroids are prednisone, deflazacort, and more recently, vamorolone. Daily or intermittent regimens of prednisone and deflazacort have shown comparable efficacy in preserving ambulation and pulmonary function, although deflazacort may offer a slightly better safety profile with less weight gain [12]. Nevertheless, long-term use of glucocorticoids is associated with notable adverse effects, including weight gain, growth suppression, behavioral changes, osteoporosis, and increased fracture risk [13,14]. Vamorolone (6 mg/kg/day) has recently been approved as an alternative to traditional glucocorticoids for the treatment of DMD. It has demonstrated a potentially improved safety profile while effectively slowing disease progression in both the short and medium term. Notably, vamorolone has been shown to improve bone turnover, promoting greater linear growth and reducing fracture risk compared to traditional glucocorticoid therapy. However, it does not eliminate other common side effects such as Cushingoid features syndrome, weight gain, or personality changes [15]. Glucocorticoid-induced growth suppression and osteoporosis are well-described in other pediatric cohorts (e.g., rheumatologic disease, cystic fibrosis). However, in DMD, additional factors—namely dystrophin deficiency-driven muscle loss that reduces mechanical bone loading and up-regulates RANKL pathways—exacerbate bone fragility beyond GC effects alone. Despite these advances, treatment remains a balance between maximizing functional benefit and minimizing side effects, necessitating individualized therapy plans for each patient.

### 3.2. Adrenal Insufficiency Secondary to Chronic Glucocorticoid Treatment and Management

Adrenal insufficiency represents a significant endocrine risk in individuals with DMD undergoing prolonged glucocorticoid therapy. Chronic steroid exposure suppresses the hypothalamic–pituitary–adrenal (HPA) axis, often resulting in inadequate cortisol production during physiological stress. In boys with DMD, this suppression can persist even after steroid discontinuation, increasing the risk of adrenal crisis during illness, trauma, or surgery. Clinical protocols recommend treating all patients receiving daily corticosteroids for more than four weeks as potentially adrenally suppressed, with stress-dose corticosteroids required during acute events [16]. Management strategies include gradual tapering schedules, availability of parenteral hydrocortisone, and training families in emergency administration. Endocrine guidelines emphasize the use of steroid emergency cards and proactive coordination between neuromuscular and endocrine teams to ensure safety in both inpatient and outpatient settings.

### 3.3. Effects of Glucorticoids on GH and Bone Growth

A well-documented consequence of glucocorticoid therapy in DMD is impaired growth. These steroids inhibit the release of growth hormone and interfere with its peripheral actions, particularly antagonizing insulin-like growth factor 1 (IGF-1) action [17]. The growth-inhibiting effects of glucocorticoids are complex and involve multiple mechanisms. One of the main harmful impacts is their toxic effect on the growth plate.

Glucocorticoids are believed to impair growth plate function by suppressing the differentiation of chondrocytes and osteoblasts, promoting apoptosis in these bone-forming cells, and by altering the local production of key paracrine factors such as insulin-like growth factor 1 (IGF-1) and C-type natriuretic peptide. Additionally, excess glucocorticoids interfere with the growth hormone (GH)–IGF-1 axis, leading to reduced GH secretion and increased GH and IGF-1 resistance at the tissue level. This is supported by a study involving glucocorticoid-treated boys with DMD where 45% of participants had subnormal stimulated GH levels (peak GH ≤ 7 ng/mL), compatible with a significant degree of GH suppression [18].

### 3.4. Causes of Short Stature in Duchenne Muscular Dystrophy (DMD) and Treatment

Short stature is a common and multifactorial manifestation in patients with DMD. Compared to the general population and to individuals with BMD, DMD patients exhibit significantly greater growth impairment. Data indicate that the average height of boys with DMD is approximately 1.08 standard deviations (SD) below the population mean, while BMD patients show a smaller reduction of 0.27 SD, suggesting a disease-intrinsic effect of dystrophin deficiency on longitudinal growth [19]. Notably, mutations affecting the Dp71 isoform of dystrophin are associated with a markedly higher prevalence of severe short stature (<−2.5 SD), reinforcing the hypothesis of a direct genetic contribution to impaired growth. This is supported by other findings showing that specific Dp71-related mutations correlate with increased growth deficiencies [19,20].

In addition to genetic mechanisms, glucocorticoid therapy remains a significant contributor to growth impairment. Long-term steroid use, especially in daily regimens, disrupts the GH–IGF-1 axis, impairs chondrocyte differentiation, and damages epiphyseal growth plates, leading to suppressed linear growth [17,18,21]. Moreover, skeletal disproportions, such as shortened limbs relative to trunk length, have been observed in glucocorticoid-treated patients, suggesting that growth abnormalities in DMD may involve primary skeletal dysregulation [22]. Collectively, these findings highlight that short stature in DMD arises from a complex interaction of genetic, structural, and pharmacological factors, rather than being a consequence of therapy only.

Given the multifactorial nature of growth failure, growth hormone (GH) therapy has been explored as a potential treatment option in DMD patients with short stature undergoing steroid treatment. In one study, GH therapy increased growth velocity from 1.3 cm/year to 5.2 cm/year after one year of treatment, in a small cohort of DMD patients [18]. Similarly, Lavi et al. [21] reported that GH therapy over 6 to 18 months improved growth rates from 0–3.25 cm/year to 3.3–7.8 cm/year, although specific time points of measurement were not detailed. Importantly, neither study observed adverse effects on cardiac or respiratory function, nor did GH therapy appear to accelerate neuromuscular decline, suggesting a favorable safety profile in this context [21]. Although GH therapy has been shown to improve growth velocity in some boys with DMD undergoing glucocorticoid therapy, the response appears to be variable, raising the hypothesis of partial GH resistance in certain individuals. Glucocorticoids are known to impair GH secretion and may also reduce GH receptor expression and signaling efficiency, potentially attenuating IGF-1-mediated anabolic effects. Moreover, chronic inflammation and increased proinflammatory cytokines such as TNF-α and IL-6 are common features of dystrophic muscles and have been shown in other muscle-wasting conditions to interfere with IGF-1 bioavailability and receptor signaling. Recent findings also point to the role of microRNAs as key mediators linking chronic inflammation to endocrine dysfunction. In particular, some microRNAs have been found to impair GH and IGF-1 signaling by targeting critical components of these pathways, contributing to growth impairment and insulin resistance in children with chronic inflammatory diseases [23]. Although this mechanism has not yet been directly demonstrated in DMD, it represents a plausible epigenetic layer contributing to hormonal resistance and warrants further investigation. However, specific studies directly demonstrating GH resistance in DMD patients are missing.

In addition to GH therapy, growth hormone secretagogues (GHS) have shown promise in preclinical models. In mdx mice, a model of DMD, the synthetic GHS compounds EP80317 and JMV2894 have been shown to enhance muscle function, reduce fibrosis, and modulate inflammation [24]. These effects were found to be independent of GHS-R1a receptor activation and of the GH/IGF-1 axis, suggesting alternative mechanisms, potentially through the inhibition of metalloproteinases involved in extracellular matrix remodeling [21,24]. While these results are encouraging, further studies are needed to validate the efficacy and safety of these agents in human subjects.

In DMD, chronic inflammation contributes significantly to the disruption of the IGF-1 axis, which is essential for muscle regeneration and anabolic signaling. Proinflammatory cytokines, particularly tumor necrosis factor-alpha (TNF-α), interfere with IGF-1 signaling through inhibitory crosstalk mechanisms. This includes impaired activation of the PI3K/Akt pathway and suppression of IGF-1 receptor function in skeletal muscle. Such molecular interactions lead to decreased protein synthesis and increased catabolism, exacerbating muscle wasting in DMD. Evidence from skeletal muscle models suggests that TNF-α can blunt the anabolic response to IGF-1, providing a pathophysiological link between chronic inflammation and endocrine dysfunction in muscular dystrophies [25] in accordance with previous published data in other chronic inflammatory diseases [23].

Delayed puberty is a frequent endocrine complication, with 82% of adolescents undergoing daily glucocorticoid therapy exhibiting delayed pubertal development, low luteinizing hormone (LH) levels, and undetectable testosterone concentrations [26]. Given these risks, careful monitoring is essential to mitigate adverse effects.

### 3.5. Pubertal Delay and Treatment

Delayed puberty is a prevalent endocrine complication in DMD, which is thought to be mainly caused by the suppression of adrenal androgens and of the hypothalamic–pituitary–gonadal axis induced by daily glucocorticoid intake, which inhibits GnRH production, reducing LH and FSH release. This could cause persistent hypogonadotropic hypogonadism [22,27]. Furthermore, other mechanisms must be considered. As previously mentioned, DMD is characterized by a persistent inflammatory response that results in elevated levels of cytokines such as TNF-α, IL-6, and IL-1. These cytokines inhibit the secretion of gonadotropin-releasing hormone (GnRH) and luteinizing hormone (LH), reducing the testicles’ sensitivity to gonadotropic hormones and lowering testosterone levels. Additionally, the delay in puberty prevents bone mass accumulation, increasing the risk of osteoporosis and fractures [22].

One clinical trial reported that 65% of patients with DMD experience delayed puberty. The serum levels of luteinizing hormone (LH) and testosterone were undetectable in boys with delayed puberty (average age of 14.3 years). Among those who underwent the luteinizing hormone-releasing hormone (LHRH) stimulation test, there was no hypothalamic–pituitary–gonadal axis activation. Additionally, the prolonged human chorionic gonadotropin (HCG) test revealed a poor testosterone response, consistent with functional central hypogonadism [26]. Testosterone replacement therapy has been explored [27]. In a retrospective study involving 16 boys aged ≥14 years (mean: 15.0 years) with DMD and pubertal delay, treatment with both oral and intramuscular testosterone led to a significant increase in bone mineral density (BMD), stabilization of vertebral fractures, and improved motor function in some non-ambulatory patients. Importantly, no significant adverse effects on cardiac or metabolic function were observed, showing a good safety profile [27]. A separate study by Wood et al. (2021) [28] evaluated the administration of intramuscular testosterone propionate in adolescents with DMD receiving glucocorticoids, aged between 12.0 and 16.9 years (mean: 13.8 years, mean bone age: 9.4 years). Over a period of two years, testosterone doses were gradually increased with administration every four weeks, according to the following schedule: 50 mg for 12 weeks, 100 mg for 40 weeks, 150 mg for 24 weeks, and finally 250 mg for 28 weeks. As a result, serum testosterone levels rose from less than 2.0 nmol/L to 9.7 nmol/L. This stimulated the hypothalamic–pituitary–gonadal axis, leading to pubertal progression (Turner G4/G5) and increased testicular volume, with an average height gain of 8.8 cm. In addition, bone mineral density in the spinal column remained stable or improved, and muscle function was maintained. The benefits of testosterone therapy extended beyond physiological changes, as assessed through the Pediatric Quality of Life Inventory (PedsQL). Following treatment, quality-of-life scores improved from 74.6 to 80.2, reflecting positive changes in body image and self-esteem. Patient interviews reported reduced psychological distress related to short stature and delayed puberty, further supporting the role of testosterone in clinical management [28]. These findings suggested that testosterone therapy should be considered as a standard intervention for pubertal delay in boys with DMD undergoing long-term glucocorticoid treatment, aged ≥12 years, according to Wood et al. [28], and ≥14 years, according to Lee et al. [27]. A further recent review by Sodero et al. provided the first consensus-based protocol for pubertal induction in DMD, recommending intramuscular testosterone starting at 50 mg monthly and titrating to 250 mg over two years, with careful monitoring of bone mineral density and psychosocial parameters. Incorporating these steps may mitigate fracture risk, optimize peak bone mass, and improve quality of life in this vulnerable population [29].

### 3.6. Obesity, Metabolic Derangements, and Management

The loss of muscle mass is related to impaired glucose metabolism and increased insulin resistance, as muscle is the primary tissue responsible for insulin-mediated glucose disposal. Chronic inflammation in skeletal muscle is a key factor contributing to insulin resistance [27]. In conditions like DMD, increased inflammatory cytokines such as IL-6 and TNF-α [22] contribute to disrupting insulin signal transduction by activating specific signaling pathways, including IKK/NF-κB and JNK. This leads to increased phosphorylation of serine residues on IRS-1 (insulin receptor substrate-1), which in turn further disrupts normal insulin signaling. Insulin resistance is reported overall in approximately 80% of DMD patients, compared to only 14% in the general population. However, one of the primary factors contributing to insulin resistance in DMD is the absence of dystrophin. Normally, insulin stimulates the recruitment of the GLUT4 transporter to the sarcolemma, enhancing glucose absorption, but the lack of dystrophin and the resulting damage to the dystrophin–glycoprotein complex disrupts the stability of the sarcolemma, leading to changes in cell signaling. Other important factors include decreased physical activity, which leads to lower energy expenditure and further decreases insulin sensitivity. Additionally, the long-term use of GC corticosteroids stimulates gluconeogenesis, encourages the accumulation of visceral fat, and reduces tissue sensitivity to insulin [17,30,31,32,33]. Insulin resistance may worsen the progression of DMD by increasing oxidative stress, inflammation, and mitochondrial damage [34].

Patients with DMD experience significant metabolic alterations, particularly as a result of prolonged corticosteroid use [17,30,31,33]. One study reported that the prevalence of obesity among children with DMD increased from 16.7% at age five to 50.6% by age eleven, with obesity being associated with an increased risk of fractures and an earlier onset of obstructive sleep apnea (OSA) [31]. Similarly, Davidson et al. [32] reported that by age ten, 50% of children with DMD were classified as obese, followed by a decline in body mass index (BMI) as ambulation was lost [32]. This decline in BMI correlated with reduced lung function, as measured by Forced Vital Capacity (FVC%). In a prospective observational study, children aged 10–13 years were found to be at the highest risk of nutritional imbalances, while the prevalence of malnutrition peaked in the 14–17-year-old age group [33]. Given these findings, early detection of malnutrition is crucial in managing DMD. The Pediatric Nutrition Screening Tool (PNST) has been proposed as a practical method for assessing nutritional risk. Regular nutritional assessments and individualized dietary interventions are key to maintaining a balance between obesity prevention in childhood and the risk of malnutrition in adolescence [33]. Further metabolic complications include hepatic dysfunction, particularly in relation to corticosteroid-induced metabolic syndrome and hepatic steatosis.

Tang et al. investigated this issue, finding that non-ambulatory patients in the late stages of the disease presented lower BMI, and significantly reduced Homeostatic Model Assessment for Insulin Resistance (HOMA-IR) and triglyceride levels compared to ambulatory patients. Hepatic steatosis, assessed via the Nakagami Parameter Index (NPI), was identified in 40.48% of DMD patients, with a notably higher prevalence (66.67%) among non-ambulatory individuals. These findings suggest that while glucocorticoid therapy in ambulatory patients is a key risk factor for hepatic steatosis, malnutrition plays a key role in hepatic dysfunction among non-ambulatory individuals [35].

Treatment with recombinant human insulin-like growth factor-1 (rhIGF-1) has been used to improve insulin sensitivity, as its receptor shares the same intracellular mediators as the insulin receptor, showing positive effects on subsequently improved carbohydrate metabolism. IGF-1 treatment reduced fasting insulin levels and improved insulin sensitivity as expected, as IGF-1 exhibits insulin-like properties. Additionally, it promoted increased linear growth and lean body mass [18].

Long-chain omega-3 fatty acids (U-3LCPUFA) have also been shown to improve insulin sensitivity and slow down muscle loss in patients with Duchenne muscular dystrophy (DMD). A daily intake of 2.9 g of omega-3 (comprising 450 mg of eicosapentaenoic acid (EPA), 2250 mg of docosahexaenoic acid (DHA), and 200 mg of other U-3 fatty acids) for 6 months resulted in a significant reduction in fasting insulin levels and the insulin resistance index (HOMA-IR). Additionally, patients who received omega-3 showed a decrease in fat mass and delayed muscle mass loss compared to those treated with a placebo [35].

Dyslipidemia is a metabolic condition found in 97% of DMD patients, regardless of obesity. Lipid abnormalities have also been observed in asymptomatic mutation carriers, suggesting an association with dystrophin deficiency, although the mechanisms are not fully understood. Additionally, the loss of muscle mass and increase in fat mass led to reduced fatty acid oxidation and increased lipid deposition. Treatment with glucocorticoids further contributes to hyperlipidemia, hyperinsulinemia, and altered lipogenesis [34]. The consumption of long-chain omega-3 fatty acids promotes fat loss. EPA and DHA have been shown to decrease lipogenesis and enhance fatty acid oxidation. This is achieved through the activation of nuclear receptors such as PPARα and PPARβ, which regulate the expression of various genes involved in fatty acid metabolism. These genes play a role in transporting fatty acids across cell membranes by binding them within cells and facilitating mitochondrial and peroxisomal beta-oxidation [36].

### 3.7. Bone Disorders and Their Management in DMD

#### 3.7.1. Skeletal Abnormalities and Fracture Risk in Duchenne Muscular Dystrophy (DMD)

Skeletal abnormalities resulting from low bone mineralization (osteopenia/osteoporosis) are a major comorbidity in DMD, significantly increasing the risk of fractures. Up to 60% of boys with DMD will experience at least one long bone fracture, most commonly in the femur, tibia, or fibula, even in the absence of glucocorticoid (GC) therapy, further increasing the risk of fractures [37]. Conversely, vertebral fractures are the most common manifestation of glucocorticoid-induced osteoporosis, occurring in more than 50% of glucocorticoid-treated boys [37,38,39]. Due to their paucisymptomatic nature, vertebral fractures often remain undiagnosed; however, if left untreated, they can lead to spinal deformities, chronic pain, and further loss of function [37,40].

Monitoring bone health in DMD is an important component of international care guidelines. Identifying bone fragility is important to inform decisions about the use of bisphosphonate drugs in the treatment and prevention of osteoporosis in these patients.

The mechanisms underlying increased bone fragility in DMD are complex and multifactorial. Key contributing factors include prolonged immobilization, glucocorticoid therapy, and muscle atrophy [41]. Despite their detrimental effects on bone health, glucocorticoids (such as prednisone and deflazacort) remain the cornerstone of DMD treatment, as they slow disease progression and prolong ambulation [37]. However, long-term glucocorticoid use impacts bone metabolism negatively by increasing RANKL expression, which enhances osteoclast activity and accelerates bone resorption [42]. Additionally, glucocorticoids reduce osteoblast lifespan and impair osteocyte signaling, further weakening bone structure [42]. Beyond their direct effects on bone cells, glucocorticoids disrupt calcium homeostasis by reducing intestinal calcium absorption and increasing renal calcium excretion, leading to hypocalcemia [42,43].

Vitamin D deficiency is another key contributor to impaired bone health in DMD. While reduced sunlight exposure due to limited mobility plays a role, the primary issue is glucocorticoid-induced suppression of vitamin D metabolism in the kidneys, which decreases the availability of active vitamin D [38,41].

Inflammation, as previously described, plays a crucial role in the progressive muscle degeneration seen in Duchenne muscular dystrophy (DMD), but it also has a significant impact on bone metabolism. As muscle fibers continuously degenerate and regenerate, chronic systemic inflammation develops, characterized by elevated levels of proinflammatory cytokines such as TNF-α, IL-1β, and IL-6, along with osteokines, like osteopontin, and myokines such as FGF-21 [41,44]. These inflammatory molecules promote osteoclast differentiation and activity, while at the same time they inhibit osteoblast function, ultimately shifting bone metabolism toward increased resorption and skeletal weakening [40]. 

In addition to inflammation, progressive muscle weakness and reduced mobility contribute significantly to bone deterioration in DMD. Bone health depends on mechanical loading, as weight-bearing activities stimulate osteoblasts and promote bone mineralization. When bones are subjected to regular loading, bone formation is enhanced, but when movement is limited, mineralization declines, and bone loss occurs [38,41]. As chronic inflammation disrupts the dystrophin-associated protein complex (DAPC), it induces structural changes that affect the stability and elasticity of the myotendinous junctions [44]. Over time, these alterations progressively reduce mechanical stimulation on the bone, impairing bone remodeling, decreasing bone strength, and accelerating bone loss [38,44].

Given the strong susceptibility to bone fragility and fractures, treating secondary osteoporosis is a critical component of the multidisciplinary management of DMD [45]. Effective bone health management can improve both quality of life and survival outcomes [43]. The goal of interventions is to slow down bone resorption and/or promote bone formation, increasing bone mineral density during corticosteroid therapy to reduce fracture risk [38]. Currently, fracture prevention focuses on vitamin D supplementation (with or without calcium) and physical activity. Bisphosphonates are widely used for osteoporosis, including glucocorticoid-induced osteoporosis, but evidence supporting their use in DMD patients is limited, except in cases with vertebral fractures or long bone fragility [44]. Bisphosphonates work by inhibiting bone resorption through osteoclast apoptosis [38]. CDC guidelines from 2018 recommend starting bisphosphonate therapy (preferably intravenously) when early signs of bone fragility appear, such as a vertebral trauma or long bone fractures. This treatment aims to preserve vertebral height, improves lung volumes, reduces pain, and prevents further fractures. The effectiveness of beginning therapy at an earlier stage in preventing new vertebral fractures is still unclear [45]. Studies disagree on which bisphosphonates should be used, on doses, and treatment duration, though intravenous administration is believed to offer the greatest benefit for vertebral fractures. These drugs are generally well tolerated, but renal function should be monitored before beginning therapy, as caution is required in patients with renal insufficiency [46]. Side effects, especially following the first infusion, are common and include fever, muscle aches, nausea, and vomiting, which can be clinically significant against the background of adrenal insufficiency in the context of long-term use of GC in this population. Rarer side effects in DMD include rhabdomyolysis [43]. Evidence supports the use of bisphosphonates to improve lumbar spine BMD Z-scores in DMD patients with glucocorticoid-induced osteoporosis, with the best results seen in those treated with intravenous zoledronate in controlled settings [45].

Testosterone therapy, used to address delayed puberty, growth impairment, and glucocorticoid-induced osteoporosis in DMD, helps increase trabecular bone density. Testosterone replacement usually starts around age 14 in accordance with the treatment schedule outlined in the designated section, with expected benefits including improved bone density, muscle strength, and energy levels [38,39]. Calcium, in combination with phosphate, is essential for bone health but can be poorly absorbed due to corticosteroid treatment. Adequate calcium intake may help slow down bone resorption and improve bone health in DMD patients, with recommended daily doses of 750 mg. Vitamin D supplementation is also important for optimal calcium absorption [38].

Bone mineral density decline in DMD becomes more evident with the loss of independent ambulation, regardless of corticosteroid use. Reduced mechanical loading impairs osteoblast function and promotes osteoclastic resorption. However, studies show that bone mineral density improves not only with calcium intake but also with regular, appropriate physical exercise [38].

#### 3.7.2. New Drugs for Bone Health in DMD

Bone remodeling is a process that ensures the maintenance of bone structure and volume, regulated by both local and systemic factors [47]. Among these, the RANK/RANKL/OPG signaling pathway plays a pivotal role in the balance between bone formation and resorption. Disruptions in this system, such as those caused by inflammation and glucocorticoids, can adversely affect bone metabolism as higher RANKL expression stimulates osteoclast activity, while lower levels of osteoprotegerin (OPG) fail to counteract this effect, leading to accelerated bone breakdown. The significance of the RANK/RANKL/OPG pathway may also extend to the pathophysiology of muscular diseases, including muscular dystrophy [42]. RANKL-targeting drugs show therapeutic potential in DMD, improving not only skeletal muscle function but also exerting anti-resorptive effects on bone. Studies in murine models of DMD treated with glucocorticoids have demonstrated that anti-RANKL therapy enhances both bone health and muscle function, suggesting that this approach may mitigate muscle damage without inducing the skeletal side effects typically associated with steroids [47].

Finally, teriparatide, a parathyroid hormone analog, is being studied for its potent osteoblast-stimulating activity [46]. It is an anabolic agent that promotes bone formation by increasing the number of osteoblasts, activating pre-existing osteoblasts, and reducing their apoptosis. The predominant effects of teriparatide therapy are observed in areas with high bone turnover, primarily trabecular bone (such as the spine), which are preferentially affected by GCs, leading to improvement in skeletal microarchitecture. This clinical study, conducted over approximately two years, demonstrated that treatment with teriparatide in six patients with DMD maintained stable bone health, with modest increases in P1NP (a marker of bone formation), and no safety concerns [48].

Givinostat is a histone deacetylase inhibitor (HDAC) that modulates the deregulated activity of histone deacetylases in dystrophic muscle. The mechanism of action of givinostat has the potential to inhibit the pathological hyperactivity of HDACs in an attempt to interfere with the cascade of events that lead to muscle damage, thus counteracting the pathophysiology of the disease and slowing muscle deterioration. This drug, in oral formulation, has been approved in the United States for patients with DMD from the age of six, with any genetic variant of the disease [49]. Ongoing clinical studies are important to evaluate the potential of HDAC inhibition and bone health in DMD. In conclusion, the primary goal is to increase bone mass that significantly impacts survival. New medications are currently being developed to restore the normal anabolic activity of bone in children with osteoporosis. The main management strategies for endocrine, metabolic, and bone disorders in DMD are summarized in Figure 3.

Endocrine disorders in DMD are complex and multifactorial and represent a significant challenge. These include adrenal insufficiency, impaired growth, delayed puberty, obesity and related comorbidities, and osteoporosis. Endocrine care requires proactive monitoring and early intervention in order to manage all these conditions appropriately. Failure to recognize endocrine diseases and inadequate therapy can lead to life-threatening emergencies.

Finally, in children with DMD, case-specific nutritional interventions are needed for malnutrition that may be related to chewing and swallowing difficulties. Enteral feeding should be considered in cases of severe swallowing difficulties or weight loss resulting from the inability to consume adequate oral nutrition. Gastrointestinal complications such as constipation, delayed gastric emptying, and gastroesophageal reflux are observed in patients with DMD. Nutritional intervention must ensure adequate caloric and nutrient intake and act on the risk of bone demineralization including adequate calcium and vitamin D intake, especially in patients treated with corticosteroids. The recommended elemental calcium intake for children aged 4 to 8 years is 1000 mg/day and increases to 1300 mg/day for children aged 9 to 18 years. After 18 years, 1000 mg/day of calcium is recommended. Food sources rich in calcium include dairy products (milk, yogurt, and cheese), green leafy vegetables (kale, broccoli), sardines and salmon, and fortified foods (some cereals, non-dairy milk substitutes, and bread). Adequate exposure to the sun converts vitamin D into an active, usable form. Some patients with limited ability to perform physical activity are at risk of becoming overweight, and prevention and a specific diet are needed in these cases. Our review notes that there is a lack of interventional studies targeting obesity or insulin resistance specifically in DMD. Nutritional counseling and moderated exercise (as tolerated) are generally recommended, as in other pediatric obesity contexts [18], but no DMD-specific weight management trial was identified. This gap suggests that formal evaluation of tailored dietary or pharmacological interventions (for example, metformin for insulin resistance) in DMD might be warranted.

Addressing these issues requires a multidisciplinary approach integrating neurological, endocrinological, and nutritional expertise (Figure 4). Much progress has been made in the knowledge and treatment of DMD. There currently are innovative strategies available that aim to provide patients with dystrophin or correct genetic mutations (such as gene therapy and exon skipping) but also “classic” strategies based on drugs, which aim to reduce inflammation, fibrosis, and muscle degeneration [50,51,52]. Cell therapy for muscular dystrophy has, unfortunately, had limited success to date, mainly due to poor engraftment of donor cells, especially in fibrotic muscle at an advanced stage of the disease [53]. CRISPR currently has great potential and is on the threshold of clinical trials; the results will be of interest [54]. The possibility of neonatal screening would allow early diagnosis and timely intervention.

Endocrine, metabolic, and bone complications are widespread and clinically significant in patients with DMD that are associated with a high burden of endocrine disturbances (such as short stature, delayed puberty, and adrenal insufficiency), metabolic derangements (notably obesity and insulin resistance), and skeletal fragility (low bone mineral density and fractures) beyond the well-recognized muscular degeneration [1,2]. These multisystem issues appear interrelated, largely stemming from the disease pathology itself and its long-term corticosteroid treatment, which together contribute to a cascade of hormonal and metabolic imbalances [3]. The findings underscore the importance of comprehensive care in DMD that addresses these domains alongside motor function. Endocrine disorders are prominently represented among the complications of DMD. Short stature is nearly universal in long-term glucocorticoid-treated patients but is observed, to a milder degree, even in steroid-naïve DMD patients [4,5]. Before starting steroid therapy, boys with DMD often present slightly reduced growth velocities in early childhood, resulting in below-average stature for age [4]. The introduction of chronic high-dose glucocorticoids (the standard of care to slow muscle degeneration) then leads to a marked growth failure due to direct suppression of growth plate cartilage and endocrine axes [5]. In this review, virtually all studies reporting on linear growth documented significant growth impairment, with many patients falling below the 3rd percentile in height during adolescence [5]. There is some debate in the literature regarding the clinical impact of short stature in DMD; while some authors have hypothesized that a smaller body size may offer functional advantages (e.g., less weight for weakened muscles to carry) [6], the prevailing view is that severe growth failure is undesirable and warrants attention for its psychosocial and health implications. Management of DMD-related short stature remains challenging. Recombinant human growth hormone (rhGH) therapy has been considered as a potential intervention for extreme short stature in DMD, but evidence of its efficacy and safety is limited [7]. No randomized controlled trials have evaluated rhGH in this patient population to date [7], and the few case series or observational reports available show different results. Moreover, theoretical concerns have been raised that accelerating growth could exacerbate muscle weakness or precipitate other complications in DMD [8]. In practice, rhGH is not routinely recommended in DMD, and clinicians should primarily focus on mitigating growth suppression by using the lowest effective glucocorticoid dose and carefully monitoring growth parameters [9]. Delayed puberty (hypogonadism) is another frequent endocrine issue. Prolonged glucocorticoid use disrupts the hypothalamic–pituitary–gonadal axis, leading to hypogonadotropic hypogonadism; consequently, most boys with DMD on steroids fail to initiate or complete puberty within the normal age range [10]. Reported prevalence of delayed puberty in glucocorticoid-treated DMD varies from 50% to 100% of patients, depending on the cohort and definitions used [10,11,12]. This delay in sexual maturation can negatively affect self-esteem, bone health, and overall quality of life. Some evidence suggests that inducing puberty with testosterone therapy can yield improvements in mood, muscle mass, and bone density in these patients [13]. Indeed, several centers empirically administer low-dose testosterone by mid-adolescence if spontaneous puberty has not occurred, with reported benefits in terms of secondary sexual characteristic development and patient well-being [13,14]. However, this review highlights that there are no controlled trials specifically assessing hormone replacement with testosterone [11]. A related concern is adrenal insufficiency resulting from chronic corticosteroid therapy. Patients with DMD on long-term daily steroids develop suppression of the adrenal glands, which can lead to life-threatening acute adrenal crises under physiologic stress (such as severe infections or surgery) [15]. The articles in our review unanimously emphasize the need for vigilance of adrenal suppression: caregivers and patients must be educated on stress-dose glucocorticoid use, and medical alert measures should be in place [15,16]. Endocrine follow-ups must include regular assessment of adrenal function (for example, through monitoring cortisol levels or stimulation tests) and provision of emergency hydrocortisone kits for at-risk patients [16]. Notably, cases of adrenal crisis have been documented in DMD patients who did not receive stress doses during acute illness, underscoring that greater awareness and preventive protocols are needed in this population [15]. In terms of metabolic complications, our review confirms that obesity and altered body composition are prevalent among boys with DMD, especially those on prolonged steroid therapy. Several studies documented that well over half of ambulatory DMD patients become overweight or obese by their early teens [17,18]. Glucocorticoids favor weight gain by increasing appetite and central fat deposition, while reduced mobility further exacerbates positive energy balance [3,18]. Excess adiposity in DMD has multifaceted consequences: it may accelerate loss of ambulation, contribute to respiratory insufficiency and sleep apnea, and predispose patients to cardiometabolic complications [17]. Thus, obesity adds to the burden of DMD and complicates its management. Beyond weight gain, DMD patients exhibit other metabolic disturbances such as insulin resistance and glucose intolerance, likely as a combined effect of glucocorticoid-induced hyperglycemia and muscle loss [19]. Cases of steroid-induced diabetes mellitus in DMD have been described, although the true prevalence of diabetes in this population remains unclear and may be under-recognized [19,20]. Dyslipidemia (elevated cholesterol and triglycerides) has also been observed in some patients [19], indicating the development of a metabolic syndrome-like profile in long-term survivors. However, due to limited longitudinal data, it is uncertain how these metabolic changes translate into adult cardiovascular risk, given that many patients historically did not survive far into adulthood. As DMD care improves and lifespans lengthen, these metabolic comorbidities could assume greater importance [20].

There is a lack of interventional studies targeting obesity or insulin resistance specifically in DMD. Nutritional counseling and moderate exercise are generally recommended [18] but no DMD-specific weight management trial was identified. Bone health complications in DMD are of notice, encompassing low bone mineral density (BMD) and a high incidence of fractures. Boys with DMD have significantly reduced BMD at the lumbar spine and in long bones [21], which is exacerbated by delayed puberty, hypogonadism, and chronic immobilization in non-ambulatory stages [22]. Fracture risk in DMD was reported to be markedly elevated [23,24]. Notably, vertebral fractures are frequently under-diagnosed in this population unless spinal imaging is performed, since patients may have difficulty communicating symptoms or may attribute chronic back pain to other causes [24]. We underscore that routine surveillance (e.g., periodic spine X-rays or densitometry) is required to detect subclinical fractures and osteoporosis in DMD [22,24]. Bisphosphonates have emerged as the mainstay of osteoporosis treatment in DMD and were examined in several of the included studies. Both intravenous bisphosphonates (such as zoledronic acid or pamidronate) and oral agents (such as alendronate) have been used [25] with significant increases in BMD Z-scores at the lumbar spine, reflecting improved bone density after 1–2 years of treatment [25]. This finding graded the evidence for bisphosphonate efficacy in DMD as high-quality with respect to BMD outcomes [26]. However, it remains uncertain whether increasing BMD via bisphosphonates translates into a significantly lower incidence of fractures. Some longitudinal studies reported fewer new vertebral fractures after bisphosphonate treatment, but the evidence overall is inconclusive due to the lack of randomized controlled data and the relatively short follow-up durations in most reports [25,27]. Patients in these studies often already had established osteoporosis or prior fractures when therapy was initiated, which makes it challenging to assess true fracture prevention as opposed to treatment of existing bone disease. In addition, differences in dosing regimens and patient ambulation status across studies make direct comparisons difficult [27]. Side effects of bisphosphonates in DMD were generally consistent with those observed in other pediatric populations, with acute-phase reactions (fever, flu-like symptoms after the first infusion) being the most commonly observed [26]. These events were usually transient and manageable with supportive care. No cases of osteonecrosis of the jaw or atypical femur fracture—rare complications of long-term bisphosphonate use—were reported in the DMD cohorts, which is reassuring given the young age of most patients [25].

Our analysis of these findings reveals several overarching themes and implications. Firstly, the multisystem nature of DMD requires a multidisciplinary approach: neurologists, endocrinologists, rehabilitation specialists, cardiologists, and nutritionists must collaborate to address the full spectrum of patient needs [3,28]. The high prevalence of endocrine and bone issues in DMD, as demonstrated by this review, highlights that clinical attention cannot be limited to muscle strength or pulmonary function alone. Early involvement of endocrinology services is advisable, for instance, to monitor growth and pubertal development from childhood and to guide timely interventions (e.g., puberty induction or adrenal crisis prevention) [14,16]. Similarly, proactive bone health management should be instituted, including adequate calcium and vitamin D intake for all patients [22,29]. Several studies evidenced that vitamin D deficiency is common in DMD and can further compromise bone integrity, suggesting that routine supplementation is a simple yet important adjunct therapy [29].

## 4. Conclusions

Endocrine, metabolic, and bone disorders are increasingly recognized as significant contributors to the overall burden of DMD, profoundly affecting patients’ quality of life and long-term outcomes. These comorbidities stem from a multifaceted interplay of genetic background, chronic inflammation, physical inactivity, and prolonged glucocorticoid use. Current management remains largely supportive and is often guided by limited clinical evidence. A coordinated, multidisciplinary approach involving neurologists, endocrinologists, dietitians, and rehabilitation specialists is essential to address these issues effectively. However, critical gaps in knowledge still exist. Additionally, the development of standardized protocols and reliable biomarkers for the early diagnosis and monitoring of endocrine and metabolic dysfunctions is urgently needed. Future research should focus on these areas to refine clinical strategies and support personalized, proactive care for patients living with DMD. Longitudinal and translational studies will be pivotal in uncovering the natural progression of these disorders and in shaping evidence-based guidelines that reflect the extended lifespan and evolving needs of this patient population.

## Figures and Tables

**Figure 1 biomedicines-13-01613-f001:**
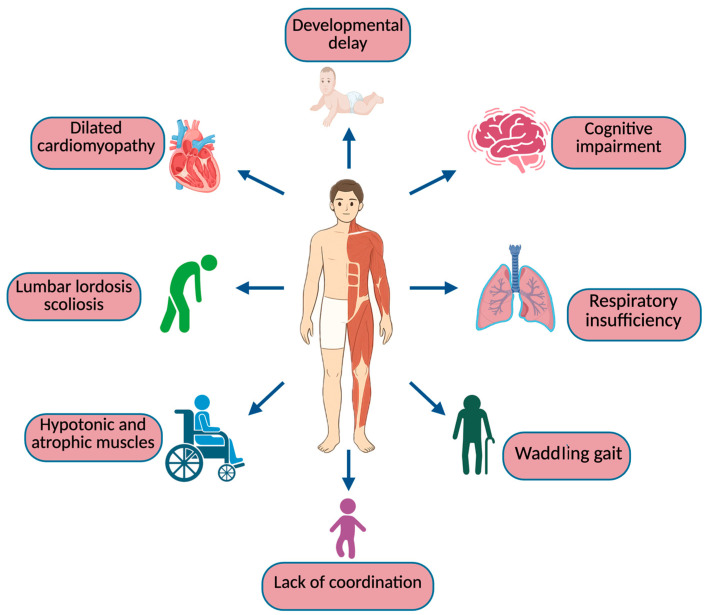
General symptoms of DMD.

**Figure 2 biomedicines-13-01613-f002:**
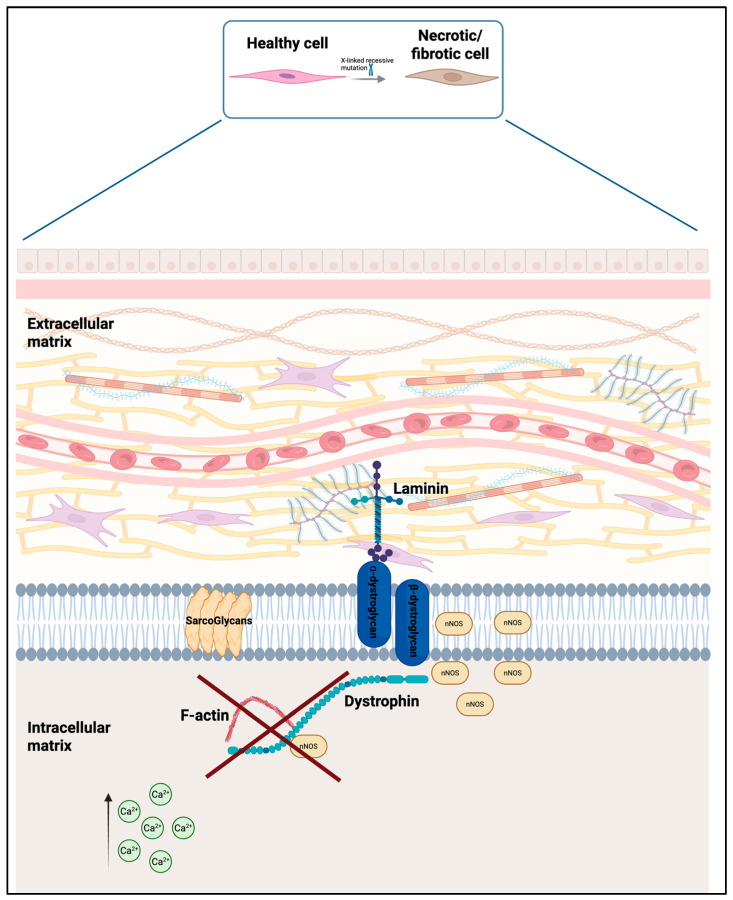
Pathophysiology of DMD. Created with Biorender.

**Figure 3 biomedicines-13-01613-f003:**
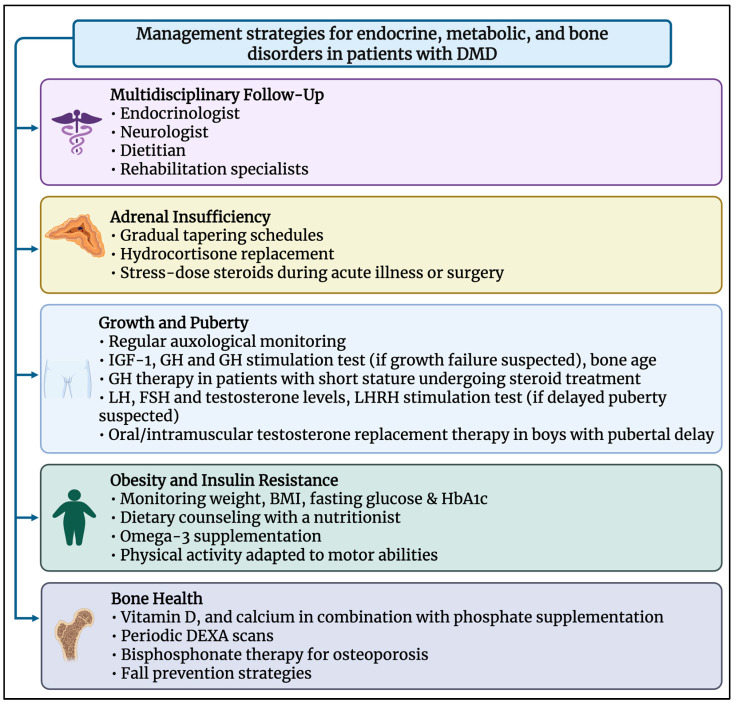
Management strategies recommended for addressing endocrine and metabolic complications in patients with DMD.

**Figure 4 biomedicines-13-01613-f004:**
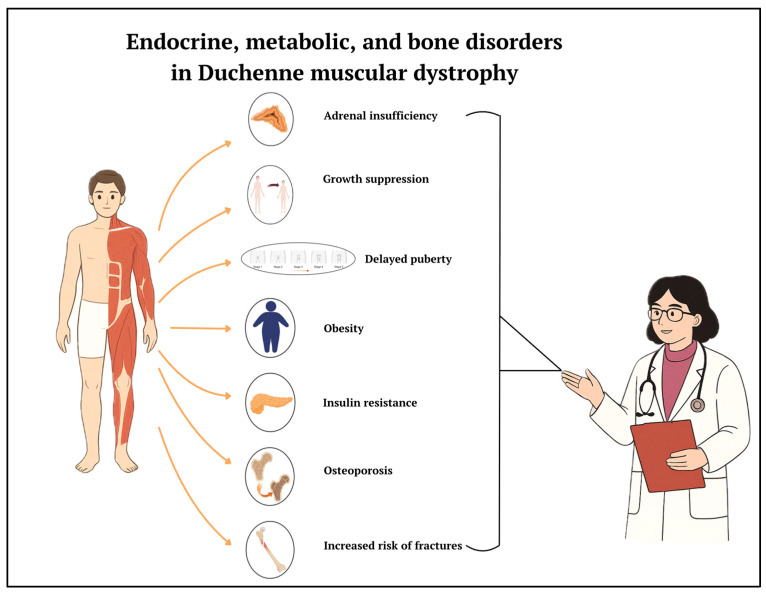
Endocrine and metabolic complications in patients with DMD.

## Data Availability

Not applicable.

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
