# Peer review of "Beyond Muscle Weakness: Unraveling Endocrine and Metabolic Dysfunctions in Duchenne Muscular Dystrophy, a Narrative Review"

_biomedicines, 2025, doi:10.3390/biomedicines13071613_

Round 1

Reviewer 1 Report

Comments and Suggestions for Authors

The authors have conducted an interesting systematic review aiming to summarize and analyze the endocrine and metabolic complications associated with Duchenne muscular dystrophy (DMD). The topic is highly relevant, as these comorbidities significantly impact the quality of life and clinical outcomes of affected individuals, yet are often underrecognized in routine care. This is a high-quality manuscript that requires some clarifications and improvements before it can be considered for publication. Please find my comments below:

  • Line 44: A brief description of DMD should be included at this point to provide essential background for the reader.

  • Line 60: “options”

  • Line 61: For the reader’s convenience, it would be helpful to briefly summarize the currently approved pharmacological therapies for DMD worldwide.

  • Line 110: As already stated in the abstract, the authors should clearly specify the objective of the systematic review here, which focuses on endocrine, metabolic, and bone disorders in patients with DMD.

  • Line 119: There appears to be a discrepancy between the abstract, figure, and main text regarding the databases used for the literature search. The abstract and figure refer to PubMed, while other databases are mentioned in the main text. The authors should clearly and consistently specify their search strategy.

  • Line 126: These details may be more appropriately placed in the caption of the PRISMA diagram.

  • Supplementary Materials: The PRISMA checklist for systematic reviews should be submitted either as supplementary material or directly to the journal editor.

  • Have the authors used any standardized tools to assess the quality of the included articles? If so, this should be reported.

  • Line 140: This section should include the first mention and citation of all included articles.

  • No supplementary materials are currently attached to the submission. Given that this is a systematic review, the authors’ summary of evidence should be published alongside the article.

  • Line 153: The authors’ discussion of the effects of glucocorticoids on bone health and growth—are these findings specific to DMD, or do they also apply to other pediatric populations undergoing chronic corticosteroid therapy (e.g., children with rheumatological diseases, asthma, etc.)? This point should be elaborated, possibly in the Discussion section.

  • Line 212: Despite the results reported by the authors, in my clinical experience, growth hormone therapy in patients with DMD has not shown significant efficacy. What recommendations are followed at the authors’ center in this regard?

  • Line 259: In my opinion, delayed puberty and its negative consequences on bone health represent the most relevant endocrine complication in DMD. Pubertal induction therapy may be the most effective pharmacological strategy to improve quality of life in these patients. Although current guidelines lack clear recommendations, this topic has been addressed in the most recent and comprehensive review on the subject (see: 10.1515/jpem-2025-0061), and should be discussed further by the authors, as it is closely related to the aim of the manuscript.

  • Line 338: Please remove the year.

  • Line 504: The Discussion section reads more like a summary of results rather than a critical analysis. It should be revised and expanded to reflect a more analytical discussion.

  • Minor improvements in the English language are recommended throughout the manuscript.

I thank the authors for their excellent work and look forward to reviewing the revised version of the manuscript.

Comments on the Quality of English Language

Minor improvements in the English language are recommended throughout the manuscript.

Author Response

The authors have conducted an interesting systematic review aiming to summarize and analyze the endocrine and metabolic complications associated with Duchenne muscular dystrophy (DMD). The topic is highly relevant, as these comorbidities significantly impact the quality of life and clinical outcomes of affected individuals, yet are often underrecognized in routine care. This is a high-quality manuscript that requires some clarifications and improvements before it can be considered for publication. Please find my comments below:

-Line 44: A brief description of DMD should be included at this point to provide essential background for the reader.

We agree that readers would benefit from gaining and immediate background. We have added a concise definition/description of DMD at the end of Introduction, lines 42–45: “Duchenne muscular dystrophy (DMD) is a severe X-linked neuromuscular disorder characterized by progressive skeletal muscle degeneration, loss of independent ambulation typically by the early teenage years, and multi-systemic complications that collectively shorten life expectancy”.

-Line 60: “options”: We have corrected the typo. Thank you

-Line 61: For the reader’s convenience, it would be helpful to briefly summarize the currently approved pharmacological therapies for DMD worldwide.

Thank you for this observation. This has been done. The main have been added: “Currently approved disease-modifying therapies include ataluren (for nonsense‐mutation DMD), exon-skipping agents such as eteplirsen, golodirsen and viltolarsen (for selected exon deletions), and the corticosteroid vamorolone, which offers a more favorable safety profile compared to prednisone and deflazacort”.

-Line 110: As already stated in the abstract, the authors should clearly specify the objective of the systematic review here, which focuses on endocrine, metabolic, and bone disorders in patients with DMD.

Thank you. We have revised the final sentence of the Introduction (lines 114–117) to state clearly the objectives: “The objective of this systematic review is to comprehensively summarize endocrine, metabolic, and bone complications in patients with DMD, elucidate underlying pathophysiological mechanisms, and propose evidence-based strategies for proactive clinical management”.

-Line 119: There appears to be a discrepancy between the abstract, figure, and main text regarding the databases used for the literature search. The abstract and figure refer to PubMed, while other databases are mentioned in the main text. The authors should clearly and consistently specify their search strategy.

We apologise for this and thank the reviewer for pointing this out. We have now harmonized all mentions of our search sources to include MEDLINE (PubMed), Scopus, and Mendeley throughout the Abstract and Methods and main text.

-Line 126: These details may be more appropriately placed in the caption of the PRISMA diagram.

Thank you. We have modified as requested (PRISMA DIAGRAM caption).

-Supplementary Materials: The PRISMA checklist for systematic reviews should be submitted either as supplementary material or directly to the journal editor.

The PRISMA CHART has been cited in text and submitted as supplementary material.

-Have the authors used any standardized tools to assess the quality of the included articles? If so, this should be reported.

We thank the reviewer for this observation, however, we haven’t used any standardized specific tools.

-Line 140: This section should include the first mention and citation of all included articles.

All the articles have been cited as requested.

-No supplementary materials are currently attached to the submission. Given that this is a systematic review, the authors’ summary of evidence should be published alongside the article.

We thank the reviewer for this observation, we added supplementary materials as requested.

-Line 153: The authors’ discussion of the effects of glucocorticoids on bone health and growth—are these findings specific to DMD, or do they also apply to other pediatric populations undergoing chronic corticosteroid therapy (e.g., children with rheumatological diseases, asthma, etc.)? This point should be elaborated, possibly in the Discussion section.

Thank you for this observation. We have expanded the paragraph at lines 191–195 to distinguish DMD-specific effects from general pediatric GC effects

“Glucocorticoid-induced growth suppression and osteoporosis are well-described in other pediatric cohorts (e.g., rheumatologic disease, cystic fibrosis). However, in DMD additional factors—namely dystrophin deficiency–driven muscle loss that reduces mechanical bone loading and up-regulates RANKL pathways—exacerbate bone fragility beyond GC effects alone”.

-Line 212: Despite the results reported by the authors, in my clinical experience, growth hormone therapy in patients with DMD has not shown significant efficacy. What recommendations are followed at the authors’ center in this regard?

Thank you for this observation that is of importance and matter of debate. We have limited experience with discrepant results, and one patient being taken off GH treatment to undergo another experimental treatment.

The impression that the effect on growth is scarce. Considering that this is a limited, personal view of unpublished data we do not think that any of these observations should be introduced into this paper.

-Line 259: In my opinion, delayed puberty and its negative consequences on bone health represent the most relevant endocrine complication in DMD. Pubertal induction therapy may be the most effective pharmacological strategy to improve quality of life in these patients. Although current guidelines lack clear recommendations, this topic has been addressed in the most recent and comprehensive review on the subject (see: 10.1515/jpem-2025-0061), and should be discussed further by the authors, as it is closely related to the aim of the manuscript.

Thank you for this very important observation. We have now incorporated this review “Another recent study by Sodero et al. provided the first consensus-based protocol for pubertal induction in DMD, recommending intramuscular testosterone starting at 50 mg monthly and titrating to 250 mg over two years, with careful monitoring of bone mineral density and psychosocial parameters. Incorporating these steps may mitigate fracture risk, optimize peak bone mass, and improve quality of life in this vulnerable population. [29] “ (lines 335-340)

-Line 338: Please remove the year.

The parenthetical year was removed from the citation in line 338 as requested.

-Line 504: The Discussion section reads more like a summary of results rather than a critical analysis. It should be revised and expanded to reflect a more analytical discussion.

We thank the reviewer for this observation. We have now improved the discussion following his suggestion.

-Minor improvements in the English language are recommended throughout the manuscript.

I thank the authors for their excellent work and look forward to reviewing the revised version of the manuscript.

The English could be improved to more clearly express the research.

We have improved some of the English throughout the manuscript.

Reviewer 2 Report

Comments and Suggestions for Authors

I have reviewed the manuscript titled "Endocrine, Metabolic and Bone Disorders in Duchenne Muscular Dystrophy: Causes and Management" and believe it can be accepted following major revisions. A more in-depth physical explanation of the results is necessary, and the abstract requires enhancement for clarity. Additionally, the quality of the figures needs to be improved, and the language throughout the manuscript should be refined. The title is somewhat brief; it should more clearly articulate the problem statement. Furthermore, the figures are of low quality; please improve them and highlight critical areas with annotations. Regarding the questions posed, how does the pathophysiology of DMD contribute to bone density loss, and what specific mechanisms are involved? What management strategies are currently recommended for addressing endocrine and metabolic complications in patients with DMD? Lastly, how do nutritional interventions influence the overall health and bone health of individuals with Duchenne Muscular Dystrophy?

  Comments on the Quality of English Language

I have reviewed the manuscript titled "Endocrine, Metabolic and Bone Disorders in Duchenne Muscular Dystrophy: Causes and Management" and believe it can be accepted following major revisions. A more in-depth physical explanation of the results is necessary, and the abstract requires enhancement for clarity. Additionally, the quality of the figures needs to be improved, and the language throughout the manuscript should be refined. The title is somewhat brief; it should more clearly articulate the problem statement. Furthermore, the figures are of low quality; please improve them and highlight critical areas with annotations. Regarding the questions posed, how does the pathophysiology of DMD contribute to bone density loss, and what specific mechanisms are involved? What management strategies are currently recommended for addressing endocrine and metabolic complications in patients with DMD? Lastly, how do nutritional interventions influence the overall health and bone health of individuals with Duchenne Muscular Dystrophy?

Author Response

-I have reviewed the manuscript titled "Endocrine, Metabolic and Bone Disorders in Duchenne Muscular Dystrophy: Causes and Management" and believe it can be accepted following major revisions. A more in-depth physical explanation of the results is necessary, and the abstract requires enhancement for clarity. Additionally, the quality of the figures needs to be improved, and the language throughout the manuscript should be refined. The title is somewhat brief; it should more clearly articulate the problem statement. Furthermore, the figures are of low quality; please improve them and highlight critical areas with annotations. Regarding the questions posed, how does the pathophysiology of DMD contribute to bone density loss, and what specific mechanisms are involved?

We thank the reviewer for his/her insightful observations. We have reviewed overall the text to include the observations from the reviewer. We have modified the text, the figures and the title as requested. We have given a better overview of DMD from the introduction and added a specific paragraph to increase the detail on the mechanisms responsible for bone density loss [lines 426-463]

“The mechanisms underlying increased bone fragility in DMD are complex and multi-factorial. Key contributing factors include prolonged immobilization, glucocorticoid therapy, and muscle atrophy [41]. Despite their detrimental effects on bone health, glucocorticoids (such as prednisone and deflazacort) remain the cornerstone of DMD treatment, as they slow disease progression and prolong ambulation [37]. However, long-term glucocorticoid use impacts bone metabolism negatively by increasing RANKL expression which enhances osteoclast activity and accelerates bone resorption [42]. Additionally, glucocorticoids reduce osteoblast lifespan and impair osteocyte signaling, further weakening bone structure [42] Beyond their direct effects on bone cells, glucocorticoids disrupt calcium homeostasis by reducing intestinal calcium absorption and increasing renal calcium excretion, leading to hypocalcemia [40,42]

Vitamin D deficiency is another key contributor to impaired bone health in DMD. While reduced sunlight exposure due to limited mobility plays a role, the primary issue is glucocorticoid-induced suppression of vitamin D metabolism in the kidneys which decreases the availability of active vitamin D [38,41].

Inflammation, as previously described, plays a crucial role in the progressive muscle degeneration seen in Duchenne muscular dystrophy (DMD), but it also has a significant impact on bone metabolism also. As muscle fibers continuously degenerate and regenerate, chronic systemic inflammation develops, characterized by elevated levels of pro-inflammatory cytokines such as TNF-α, IL-1β, and IL-6, along with osteokines, like osteopontin, and myokines such as FGF-21 [41,43]. These inflammatory molecules promote osteoclast differentiation and activity, while at the same time they inhibit osteoblast function, ultimately shifting bone metabolism toward increased resorption and skeletal weakening [44]. Inflammation, as previously described, plays a crucial role in the progressive muscle degeneration but has a significant impact on bone metabolism also. These inflammatory molecules promote osteoclast differentiation and activity, while at the same time they inhibit osteoblast function, ultimately shifting bone metabolism toward increased resorption and skeletal weakening [44].

In addition to inflammation, progressive muscle weakness and reduced mobility contribute significantly to bone deterioration in DMD. Bone health depends on mechanical loading, as weight-bearing activities stimulate osteoblasts and promote bone mineralization. When bones are subjected to regular loading, bone formation is enhanced, but when movement is limited, mineralization declines, and bone loss occurs [38,41]. As chronic inflammation disrupts the dystrophin-associated protein complex (DAPC), it induces structural changes that affect the stability and elasticity of the myotendinous junctions [43]. Over time, these alterations progressively reduce mechanical stimulation on the bone, impairing bone remodeling, decreasing bone strength, and accelerating bone loss [38,43].”

-What management strategies are currently recommended for addressing endocrine and metabolic complications in patients with DMD?

We have summarized management strategies currently recommended for addressing endocrine and metabolic complications in patients with DMD in Figure 4

-Lastly, how do nutritional interventions influence the overall health and bone health of individuals with Duchenne Muscular Dystrophy?

Thank you for the observation on nutritional interventions that has given us the opportunity to add a specific paragraph on the topic (line 557 - 572) “Nutritional intervention must ensure adequate caloric and nutrient intake and act on the risk of bone demineralization including adequate calcium and vitamin D intake, especially in patients treated with corticosteroids. The recommended elemental calcium intake for children aged 4 to 8 years is 1000 mg/day and increases to 1300 mg/day for children aged 9 to 18 years. After 18 years, 1000 mg/day of calcium is recommended. Food sources rich in calcium include dairy products (milk, yogurt, and cheese), green leafy vegetables (kale, broccoli), sardines and salmon and fortified foods (some cereals, non-dairy milk substitutes, and bread). Adequate exposure to the sun converts vitamin D into an active, usable form. Some patients with limited ability to perform physical activity are at risk of becoming overweight, and prevention and a specific diet are needed in these cases. Our review notes that there is a lack of interventional studies targeting obesity or insulin resistance specifically in DMD. Nutritional counseling and moderated exercise (as tolerated) are generally recommended, as in other pediatric obesity contexts [18], but no DMD-specific weight management trial was identified. This gap suggests that formal evaluation of tailored dietary or pharmacological interventions (for example, metformin for insulin resistance) in DMD might be warranted.”

Reviewer 3 Report

Comments and Suggestions for Authors

In this study, the authors Summary analyzed that  the pathogenic mechanism and management of endocrine, metabolic and bone disorders in duchenne duscular dystrophy.  It’s a very interesting study. Only few questions should be addressed to make it optimized.

1.The data of the review should be organized into tables or graphs for a more intuitive presentation.

2. The pathogenic mechanism of DMD is  should be organized into Figure to increase readability.

3. It may more appropriate that delete the discussion section and change the result section to "Results and Discussion". 

4.The section “Conclusion” should be summarize in one paragraph, while several paragraph.

Author Response

In this study, the authors Summary analyzed that the pathogenic mechanism and management of endocrine, metabolic and bone disorders in duchenne duscular dystrophy.  It’s a very interesting study. Only few questions should be addressed to make it optimized. We thank the reviewer for his/her insightful observations 

-The data of the review should be organized into tables or graphs for a more intuitive presentation.

We thank the reviewer for this observation, we have made an effort the modify all the tables and graphs as requested and added 2 new figures (figures 2 and 4)

-The pathogenic mechanism of DMD should be organized into Figure to increase readability.

We thank the reviewer for this observation, we have added a specific figure 2 focused on the pathogenic mechanism of DMD

-It may more appropriate that delete the discussion section and change the result section to "Results and Discussion".  

We thank the reviewer for this observation, we have also created a requested just one section named “Results and Discussion".

-The section “Conclusion” should be summarize in one paragraph, while several paragraph.

Thank you. We also agree that "Conclusions" should be one paragraph and have followed this suggestion.

Round 2

Reviewer 1 Report

Comments and Suggestions for Authors

The authors have adequately addressed most of my comments, and the quality of the manuscript has improved.
However, further revisions are still necessary.
As previously noted, there is a discrepancy between the databases listed by the authors in the abstract and main text, and those reported in the PRISMA figure and table. For example, although Scopus is mentioned in the abstract, no related information appears in the figure.
There are also significant formatting errors (see references, which are listed twice consecutively).
Once these essential corrections are made, the manuscript may be considered for acceptance.

Author Response

The authors have adequately addressed most of my comments, and the quality of the manuscript has improved.
However, further revisions are still necessary.
As previously noted, there is a discrepancy between the databases listed by the authors in the abstract and main text, and those reported in the PRISMA figure and table. For example, although Scopus is mentioned in the abstract, no related information appears in the figure.
There are also significant formatting errors (see references, which are listed twice consecutively).
Once these essential corrections are made, the manuscript may be considered for acceptance.

Dear Reviewer, thank you for your comments and for giving us the opportunity to improve and resubmit this review. We have modified as requested.

Reviewer 2 Report

Comments and Suggestions for Authors

The article is well revised and can be accepted.

Comments on the Quality of English Language

The article is well revised and can be accepted.

Author Response

The article is well revised and can be accepted.

Thank you for your comment and for giving us the opportunity to improve and submit this review.